# A Recognition Model Incorporating Geometric Relationships of Ship Components

Shengqin Ma [1,2,3] , Wenzhi Wang [1,2], Zongxu Pan [1,2,3] , Yuxin Hu [1,2], Guangyao Zhou [1,2]
and Qiantong Wang [1,2,*]

1. Aerospace Information Research Institute, Chinese Academy of Sciences, Beijing 100094, China; mashengqin17@mails.ucas.ac.cn (S.M.); wangzw@aircas.ac.cn (W.W.); zxpan@mail.ie.ac.cn (Z.P.); yxhu@mail.ie.ac.cn (Y.H.); zhougy@aircas.ac.cn (G.Z.)
2. Key Laboratory of Technology in Geo-Spatial Information Processing and Application System, Chinese Academy of Sciences, Beijing 100190, China
3. School of Electronic, Electrical and Communication Engineering, University of Chinese Academy of Sciences, Beijing 100049, China
* Correspondence: wangqiantong23@mails.ucas.ac.cn

**Abstract:** Ship recognition with optical remote sensing images is currently widely used in fishery management, ship traffic surveillance, and maritime warfare. However, it currently faces two major challenges: recognizing rotated targets and achieving fine-grained recognition. To address these challenges, this paper presents a new model called Related-YOLO. This model utilizes the mechanisms of relational attention to stress positional relationships between the components of a ship, extracting key features more accurately. Furthermore, it introduces a hierarchical clustering algorithm to implement adaptive anchor boxes. To tackle the issue of detecting multiple targets at different scales, a small target detection head is added. Additionally, the model employs deformable convolution to extract the features of targets with diverse shapes. To evaluate the performance of the proposed model, a new dataset named FGWC-18 is established, specifically designed for fine-grained warship recognition. Experimental results demonstrate the excellent performance of the model on this dataset and two other public datasets, namely FGSC-23 and FGSCR-42. In summary, our model offers a new route to solve the challenging issues of detecting rotating targets and fine-grained recognition with remote sensing images, which provides a reliable foundation for the application of remote sensing images in a wide range of fields.

**Keywords:** optical remote sensing; rotated ship recognition; fine-grained ship dataset

## 1. Introduction

Ship recognition based on optical remote sensing images has diverse applications in fishery management, ship traffic surveillance, and maritime warfare [1]. It has emerged as a significant research field within the field of computer vision. Traditional methods for ship detection relied on geometric elements and artificially designed features to identify ships [2–4]. In recent years, advancements in Earth observation technology have significantly enhanced the acquisition capabilities of high-resolution visible remote sensing images [5]. The inherent interpretability of optical remote sensing images, coupled with their clear and rich texture information, has opened up new avenues for ship target detection and identification techniques. Given high-resolution optical remote sensing images, convolutional neural networks (CNNs) can extract more semantic information than traditional methods, and are also more robust and generalized, thus giving better detection results [6]. Among them, the coarse-to-fine network structure has excellent results in high-resolution remote sensing images, and it improves the ship detection accuracy by iteratively refining the detection results at different scales, which solves the problem of multi-scale in remote sensing images [7–16].

However, remote sensing image ship detection and recognition face challenges arising from the unique shooting angles, scene complexity, and variations in object proportions within a single image [15]. Currently, the field of ship detection and identification based on remote sensing images primarily encounters difficulties in the two following aspects: rotating the ship detection and fine-grained ship detection.

1. The ship recognition methods based on horizontal bounding boxes (HBBs) aim to produce satisfactory results. Meanwhile, methods based on oriented bounding boxes (OBBs) may require more computational power and time. Most ships in remote sensing images exhibit arbitrary orientation, a high aspect ratio, and dense distribution. Traditional HBB-based methods struggle to accurately capture the true shape of the object, often including background pixels and neighboring targets within the detection box. Additionally, when dealing with high aspect ratios, even slight angular deviations between the predicted frame and the ground truth frame can lead to a rapid decline in the intersection over union (IoU) ratio. This ratio serves as the basis for evaluation metrics and sensitive hyperparameters in non-maximum suppression (NMS).

To further enhance the ship detection accuracy in optical remote sensing images, researchers have proposed target detection methods for arbitrary directions, including both one-stage and two-stage approaches. On the other hand, the two-stage method involves extracting the candidate frames and subsequently classifying and regressing each candidate frame to obtain precise target location and category information [17–20]. Included among these, the ROI Transformer [19] solved the problem of object detection in aerial images, specifically addressing the challenges presented by the bird's-eye view perspective, highly complex backgrounds, and variant appearances of object. It applies spatial transformations on region of interest (RoIs) and learns transformation parameters under the supervision of oriented bounding box (OBB) annotations, allowing for the detection of oriented and densely packed objects. The proposed method avoids the need for a large number of anchors designed for oriented object detection and is a significant improvement over previous approaches by enabling more accurate object localization and feature extraction. On the other hand, single-stage detectors integrate feature extraction, proposal generation, and subsequent detection steps into a unified network [9,21–25]. The establishment of the rotating boxes introduces additional angular regression. Since angles are periodic, regression at some particular angles will generate abrupt changes in the loss values, interfering with the detection results of the network. Based on this problem, the circular smooth label (CSL) [21] is proposed, which converts the angular regression approach to a classification approach, since the classification results are limited and do not occur outside the defined range. Converting a regression problem into a classification problem is actually a continuous to discrete problem, and the accuracy is lost in this conversion process. By calculating the maximum loss of accuracy and the average loss, it is concluded that it is feasible to convert the angle prediction approach into a classification problem. The essential problem of rotating target detection is that the detection performance of high-aspect-ratio objects, such as ships, was lower than the overall average [26]. Currently, the mainstream strategy to solve this problem is to increase the aspect ratio range of the anchor, but this brings a greater amount of calculation and cannot fundamentally solve the problem.

2. The fine-grained recognition method for optical remote sensing ship images faces serious challenges. Ships in remote sensing images exhibit diverse types and complexities. Some classes of ships have significant appearance variations within the class. Additionally, certain ship classes with similar usage may share inter-class similarities.

One common approach to address this challenge is to employ attention mechanisms that focus on the discriminative parts of low-discriminative objects to aid recognition. For example, ref. [27] proposed an interpretable attention network for fine-grained ship classification using remote sensing images. Similarly, ref. [28] introduced a multi-source regional attention network for fine-grained object recognition in remotely sensed images. This method combines multiple information sources and employs attention mechanisms to

recognize fine-grained objects in remote sensing images. Although these methods have introduced new mechanisms for fine-grained target recognition and achieved promising results, the extracted features for fine-grained targets may lack sufficient differentiation, making an accurate distinction challenging. Further research and development are necessary to improve the accuracy and reliability of fine-grained ship recognition in remote sensing imagery.

This paper introduces a ship recognition approach recognizing rotated targets and achieving a fine-grained recognition. The proposed method extends traditional ship detection by incorporating the ship part detection. By simultaneously detecting the ship components and the whole ship, the model leverages the geometric regularity of ship components to extract the key discriminative features more efficiently. The significance of this approach includes:

1.  Generalized ship component samples. By selecting commonly found ship components such as the main gun, chimney, vertical launch system, and flight deck, the samples become more generalized. Most ships possess some or all of these components, reducing the need for extensive ship-specific data.
2.  The simplified labeling and orientation determination. Ship components often have regular shapes like rounds or squares, with symmetry. This characteristic leads to less background area for labeling and prediction while using horizontal frames. Furthermore, identifying ship components can assist in determining the actual orientation of the ship.
3.  The utilization of regular geometric relationships. While ships of different types may share similarities in their outward appearance, the distribution and arrangement of their components exhibit variations. Moreover, there are distinct characteristics in terms of the relative positions and orientations of ship components and the alignment and integration of ship components within the ship's structure. Exploiting these inherent geometric relationships can significantly contribute to the processes of detection and identification.

The ship recognition method incorporating ship components introduces a new idea and solves the problem of difficult fitting of rotating frame detection. It also introduces more detection content, contributing to the fine-grained detection of ships, and thus enhancing the scope of ship detection applications, which is significant in fishery management, ship traffic surveillance, and maritime warfare. But, it also presents new difficulties and challenges:

1.  Smaller and more challenging detection. Ship components are typically smaller than the whole ship in scale, making their detection more difficult. The reduced size increases the complexity of accurately localizing and recognizing these components.
2.  Uneven sample distribution. The number of available samples for different ship components may vary significantly, resulting in imbalanced data distributions. This imbalance can pose challenges during training and may impact the performance of the recognition model.
3.  Varied difficulty in feature extraction. Different ship components may exhibit varying levels of difficulty in feature extraction. Some components may possess distinctive features, while others may lack clear discriminative characteristics. Addressing these differences in feature extraction complexity is crucial for achieving accurate recognition across all ship components.

This paper proposes a new network structure called Related-YOLO to realize the proposed ship recognition method and overcome the aforementioned difficulties. Additionally, a new dataset is constructed to account for the wide variety of warships with interclass differences and intraclass similarities. The main contributions of this paper are as follows:

1.  Geometric relationship constraints and attention mechanism. The paper incorporates the geometric relationship constraints of ship components. This information is utilized to enhance the accuracy and reduce the false alarm rate. Additionally, an attention

mechanism is employed to weight the extracted sample features. This mechanism effectively focuses on the most relevant features and improves the overall detection accuracy.

2. Adaptive anchor box generation. A new hierarchical clustering approach is introduced to generate adaptive anchor boxes. This approach helps improve the network's ability to detect multi-scale samples. By dynamically generating anchor boxes based on the clustering algorithm, the network becomes more effective in handling objects of varying sizes.

3. Small-target detection layer. A novel network structure that incorporates a small target detection layer is designed. This layer specifically focuses on enhancing the network's capability to detect tiny targets. Small targets often pose challenges due to their limited visual information, and the inclusion of this layer aims to improve the detection performance for such cases.

4. Deformable convolution for feature extraction. This paper utilizes deformable convolution for the feature extraction of input samples. Deformable convolution allows the network to effectively extract features from samples with different shapes. By adapting the convolutional filters to the specific spatial locations, the network becomes more flexible in capturing the informative features from ship images.

## 2. Materials and Methods

The workflow of this study is shown in Figure 1. In the beginning, the detection network needs to be trained. We first input the training image and its annotation, obtain the extracted features through the backbone, and then calculate the geometric position relationship between the targets to weight the extracted features. The weighted features complete feature fusion in the neck part, and finally complete label matching and loss function calculation in the head part, and obtain the geometric relationship constraints between the targets. In the testing and inference stages, after we input the image to be detected, we obtained the final result of ship recognition through the detection network and geometric relationship constraints.

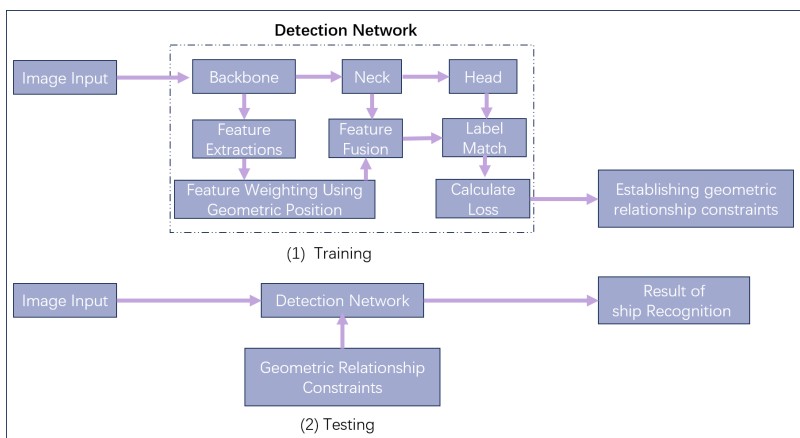

**Figure 1.** The workflow of our proposed method.

In this paper, a new neural network called related-YOLO is developed to improve ship detection by incorporating the geometric relationships of ship components in remote sensing images. The network structure is shown in Figure 2, which consists of four main components: input, backbone, neck, and head. The Related-YOLO network builds upon the YOLO v5 [29] algorithm, enhancing it with adaptive anchor boxes, deformable convolution, and the relational attention module. These modifications enable the better detection of ship components in remote sensing images, particularly in terms of small and irregular objects, and improve the overall accuracy of the ship recognition system.

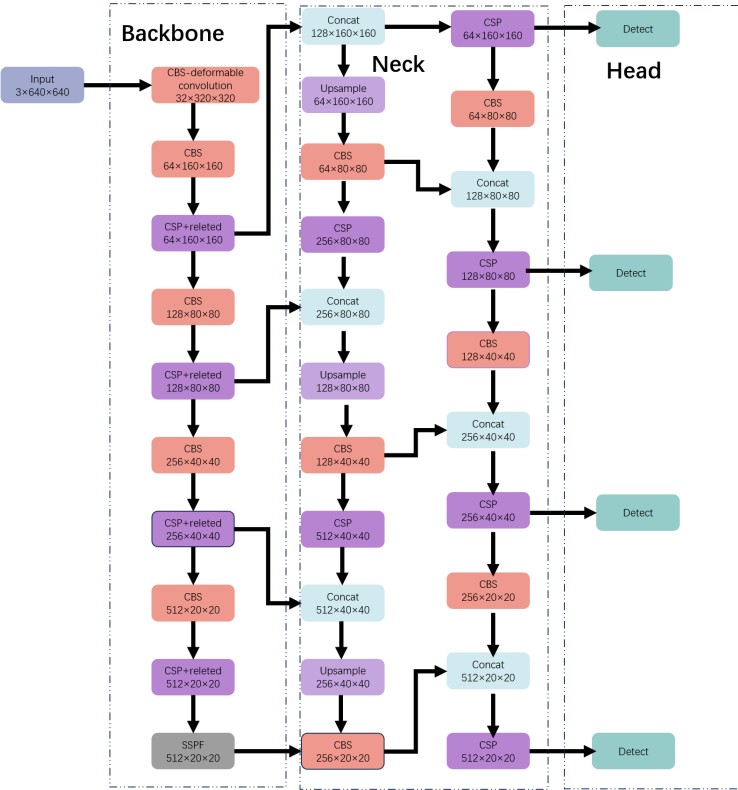

**Figure 2.** Network structure.

Data augmentation techniques are first applied to enhance the training data including ship components and ships. Additionally, the adaptive anchor boxes are generated using a hierarchical clustering algorithm. Thus, boxes are better suited for detecting small and irregular objects, such as ship components.

The backbone is responsible for downsampling the input image and extracting multi-scale features. We innovatively modified the network structure of the backbone by adding a small target detection layer to enhance the multi-scale capability of the network. Additionally, the convolutional layer of the backbone is improved by incorporating deformable convolution. This enhances the feature extraction capability, particularly for irregular targets like the main gun of the warship. Another notable improvement in the Related-YOLO network is the addition of the relational attention module at the tail of the backbone. This module introduces geometric relational constraints, leveraging the relationships between ship components. By incorporating these constraints, the network achieves a significantly improved detection accuracy.

The neck fuses the extracted features from different scales to generate semantic features. This fusion process helps capture more comprehensive information about the ship components and improves the overall understanding of the image.

At the Head of the network, the positional regression and classification are performed based on the predictive feature mappings generated in the neck section. This process outputs detection boxes for each object of interest, i.e., the ship and ship components, in the image.

## 2.1. A Hierarchical Clustering Algorithm to Implement Adaptive Anchor Boxes

In the detection network, the anchor-based mechanism generates a set of anchor boxes that serve as reference templates for target classification and bounding box regression [17]. This mechanism improves the recall ability of the network, especially for detecting small targets. However, setting the hyperparameters for anchor generation requires prior knowledge and can be challenging [30].

To address this issue, adaptive anchor box generation is introduced, which eliminates the need for manual hyperparameter tuning. In the original YOLO v5 framework [29], the adaptive anchor boxes are generated using the K-means clustering algorithm. The algorithm works as follows:

Step 1: Randomly select K initial center values.

$$\mu_1, \mu_2, \ldots, \mu_k \in \mathbb{R}^n \tag{1}$$

Step 2: Group the samples into clusters based on the distance between each sample and the center value.

$$c^{(i)} = \arg\min_j \left\| x^{(i)} - \mu_j \right\|^2. \tag{2}$$

Step 3: Update the center value of each cluster by calculating the mean of the values contained in the cluster.

$$u_j := \frac{\sum_{i=1}^m 1\left\{c^{(i)} = j\right\} x^{(i)}}{\sum_{i=1}^m 1\left\{c^{(i)} = j\right\}}. \tag{3}$$

Repeat steps 2 and 3 until the positions of the cluster centers no longer change.

The K-means algorithm, although capable of solving the problem of generating adaptive anchor boxes, has some problems in our study:

1. The number of clusters needs to be specified in advance: for the task of ship detection based on ship parts in remote sensing imagery, the size distribution of the samples is not uniform, and selecting the appropriate number of clusters may be difficult without a priori knowledge.
2. Sensitivity to initial centroid selection: different initial centroid selections may lead to different clustering results. A poor initial selection may cause the algorithm to fall into a local optimum solution.
3. Sensitive to outliers: outliers may disturb the clustering results, causing some clusters to be affected or even split into multiple clusters.

As a result, the K-means clustering algorithm is not fully applicable to our kind of multi-target and multi-size tasks with large-scale differences.

To solve these problems, this paper uses a hierarchical clustering algorithm [31]. This algorithm connects objects to form clusters based on distance. The clusters can be roughly described by the maximum distance required to connect the components. Different clusters are formed at different distances, which can be presented using a dendrogram in Figure 3. Distinguishing from the K-means algorithm, hierarchical clustering does not require the number of clusters to be pre-specified, it automatically constructs the cluster hierarchy; and it does not require an explicit initial selection—it constructs the cluster hierarchy by gradually merging or splitting data points; in addition to this, hierarchical clustering can be used to detect outliers by treating them as separate clusters, or by characterizing the cluster hierarchy according to outlier detection and elimination.

The algorithm connects objects to form clusters based on distance. The clusters can be roughly described by the maximum distance required to connect the components. At different distances, different clusters are formed, which can be presented using a tree diagram. The specific algorithm steps are:

Step 1. Initially each sample is a cluster and the distance matrix $D$ is computed:

$$D_{ij}^x = \left\| x_i - x_j \right\|_2^2 \tag{4}$$

iterate through the distance matrix $D$ to find out the minimum distance in it except the diagonal and obtain the number of the two clusters having the minimum distance, merging the two clusters into a new cluster and updating the distance matrix $D$.

Step 3. The merging process of clustering is repeated until all objects finally satisfy the number of clusters set by the termination condition.

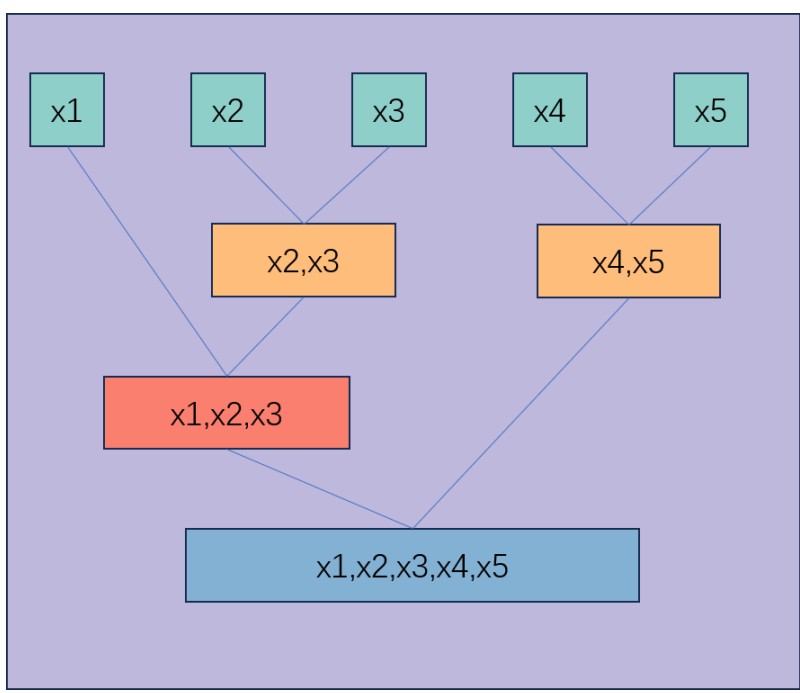

**Figure 3.** Schematic of hierarchical clustering.

### 2.2. Deformable Convolution

In detection networks, convolutional kernels are commonly used to extract features from the input data. However, traditional convolutional kernels have fixed shapes and sizes, which restrict their adaptability and generalizability when dealing with targets of varying shapes [6]. To address this limitation, deformable convolution [32] is introduced. It is shown in the Figure 4.

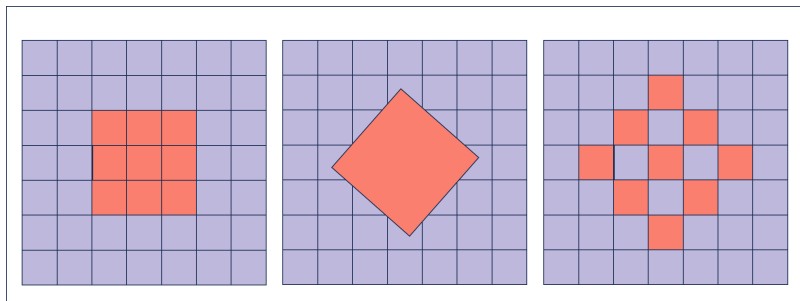

**Figure 4.** Comparison of conventional convolution and deformable convolution.

Deformable convolution extends the capabilities of traditional convolutional kernels by incorporating learnable offsets within the receptive field. These offsets are parameters that can be adjusted during training through backpropagation. By introducing these offsets, deformable convolution allows the receptive field to be dynamically adjusted and aligned with the actual shape of the target.

During the forward pass of deformable convolution, the offsets are learned and used to sample input features in a grid-like manner. This sampling process enables the receptive field to be deformed and adapt to the shape of the target being detected. As a result, deformable convolution can capture more fine-grained details and better represent the spatial characteristics of objects with irregular shapes.

The learning of offsets in deformable convolution is performed jointly with the other network parameters during backpropagation. This means that the network can learn to adjust the offsets to align the receptive field with the target shape which maximizes the detection performance.

By introducing deformable convolution, the detection network becomes more flexible and capable of handling targets with varying shapes. It improves the network's ability to capture object details and enhances the generalization of the model to different shape variations within the target class.

Overall, deformable convolution addresses the limitations of traditional fixed-shape convolutional kernels by introducing learnable offsets that allow the receptive field to be deformed and aligned with the actual shape of the target. This adaptive mechanism improves the detection network's ability to handle objects with different shapes, enhancing performance in shape-sensitive tasks like object detection.

The traditional convolution operation can be represented by Equation (1), where $p_0$ represents each point in the output feature map corresponding to the center point of the convolution kernel, and $p_n$ represents the offset of $p_0$ within the range of the convolution kernel [6].

$$y(p_0) = \sum_{p_n \in \mathcal{R}} w(p_n) \cdot x(p_0 + p_n) \tag{5}$$

In deformable convolution, an offset is introduced for each point in the output feature map. This offset is generated from the input feature map using another convolution operation, typically resulting in decimal values. The convolution kernel used to obtain the offsets is the same as a regular convolution kernel. The dimensions of the output offsets are the same as the dimensions of the input feature map, and the channel dimension is 2N, where N represents the number of offsets. These two convolutional kernels are learned simultaneously using the backward propagation algorithm with bilinear interpolation.

$$y(p_0) = \sum_{p_n \in \mathcal{R}} w(p_n) \cdot x(p_0 + p_n + \Delta p_n) \tag{6}$$

The regression process for the offsets is as follows:

Initially, a feature map is obtained by applying a regular convolutional kernel to the input image, similarly to a normal convolutional neural network. The obtained feature map is then fed into another convolutional layer to regress the offsets for deformable convolution. These offsets represent the deformation of the receptive field and are learned through backpropagation. The dimension of the offset layer is 2N because we need to account for translations in both the x and y directions. During training, both the convolutional kernel used for generating the output features and the convolutional kernel used for generating the offsets are learned simultaneously. Since the positions after adding the offsets are non-integer and do not correspond to actual pixel points on the feature map, interpolation is required to obtain the offset pixel values. Bilinear interpolation is commonly used for this purpose.

The process of bilinear interpolation can be expressed by the following equation:

$$\begin{aligned} x(\mathrm{p}) &= \sum_{\mathrm{q}} G(\mathrm{q}, \mathrm{p}) \cdot x(\mathrm{q}) \\ &= \sum_{\mathrm{q}} g(\mathrm{q_x}, \mathrm{p_x}) \cdot g(\mathrm{q_y}, \mathrm{p_y}) \cdot x(\mathrm{q}) \\ &= \sum_{\mathrm{q}} \max(0, 1 - |\mathrm{q_x} - \mathrm{p_x}|) \cdot \max(0, 1 - |\mathrm{q_y} - \mathrm{p_y}|) \cdot x(\mathrm{q}) \end{aligned} \tag{7}$$

where $f(x, y)$ represents the pixel value at the interpolated position $(x, y)$, $p(i, j)$ represents the pixel value at the nearest existing pixel points in the feature map, and the summation is performed over the four neighboring pixel points. The weights for each point are determined based on their distances to the interpolation position's vertical and horizontal

coordinates. The final term, max (0, 1, . . . ), ensures that the interpolated point does not deviate by more than one pixel from the neighboring points.

In summary, deformable convolution introduces learnable offsets to adapt the receptive field, and these offsets are generated from the input feature map using convolutional operations. The regression of offsets is performed by applying a separate convolutional layer. Bilinear interpolation is used to obtain the pixel values for the offset positions. This mechanism allows the receptive field to deform and align with the target shape, improving the adaptability of the network in capturing fine-grained details and handling objects with varying shapes.

*2.3. Mechanisms of Relational Attention*

This paper introduces geometric constraints into the network structure to improve the performance of ship component detection algorithms. These geometric constraints are incorporated using a relational attention module [33].

In this paper, the relational attention module utilizes the self-attention mechanism to model the regular geometric relationships between ship components. Each target is considered to have two features: geometric features ($f_G$) and appearance features ($f_A$), which they obtain using the self-attention mechanism. The geometric features capture the relationships between targets, while the appearance features represent the visual appearance of each target.

The self-attention [34] mechanism is a common attention mechanism used in many models. The core formula of self-attention is

$$\text{Softmax}\left(\frac{QK^\top}{\sqrt{d_k}}\right)V \tag{8}$$

where $Q$ represents the query, $K$ represents the key, and $V$ represents the value.

In the self-attention mechanism, $Q$, $K$, and $V$ are linearly transformed from the same input matrix $X$. The input matrix $X$ is the same as the input feature map. The linear transformations are defined as follows:

$$Q = XW^Q \tag{9}$$
$$K = XW^K \tag{10}$$
$$V = XW^V \tag{11}$$

Here, $W^Q, W^K$, and $W^V$ are trainable parameter matrices.

The self-attention mechanism calculates the correlation between every two input vectors using $Q$ and $K(QK^T)$. The resulting similarity matrix is then divided by $sqrt(d_k)$ to stabilize the gradient during training. After normalization using softmax, each value becomes a weight coefficient between 0 and 1, summing to 1. This weight matrix is then multiplied with $V$ to compute a weighted summation.

To handle the translation and scale transformations, a normalized four-dimensional geometric feature $(f_G^m, f_G^n)$ is used to represent the relative geometric relationship between targets m and n. This geometric feature includes the logarithm of the relative distances and sizes of the targets.

$$(f_G^m, f_G^n) = \left(\log\left(\frac{|x_m - x_n|}{w_m}\right), \log\left(\frac{|y_m - y_n|}{h_m}\right), \log\left(\frac{w_n}{w_m}\right), \log\left(\frac{h_n}{h_m}\right)\right)^T \tag{12}$$

Using the geometric features, a geometric weight $(w_G^{mn})$ is computed to constrain the geometric relationship between the different targets. The geometric features are transformed by a matrix $W_G$, and the resulting weights are pruned at 0, indicating that the geometric weights are only applied to pairs of targets with specific geometric relationships.

$$\omega_G^{mn} = \max\{0, W_G \cdot (f_G^m, f_G^n)\} \tag{13}$$

Similarly, the appearance features $(f_A^m, f_A^n)$ are used to compute an appearance weight $(w_A^{mn})$. This is achieved using a dot product between the appearance features and a trainable matrix $(W_K)$, and applying softmax to obtain the weights.

$$\omega_A^{mn} = \frac{\mathrm{dot}\left(W_K f_A^m, W_Q f_A^n\right)}{\sqrt{d_k}} \tag{14}$$

By incorporating the geometric weights and appearance weights into the network, the relational attention module considers both the geometric relationships and appearance information of the ship components, enhancing the detection performance and reducing false alarms by combining them with the NMS algorithm.

$$\omega^{mn} = \frac{\omega_G^{mn} \cdot \exp\left(\omega_A^{mn}\right)}{\sum_k \omega_G^{kn} \cdot \exp\left(\omega_A^{kn}\right)}. \tag{15}$$

And, it is possible to compute the relational characteristics of each target with respect to the other targets $f_R(n)$.

$$f_R(n) = \sum_m \omega^{mn} \cdot (W_V \cdot f_A^m) \tag{16}$$

Finally, the appearance features of the input target are augmented by summation:

$$f_A^n = f_A^n + \mathrm{Concat}\left[f_R^1(n), \ldots, f_R^{N_r}(n)\right], \text{ for all } n \tag{17}$$

### 2.4. Datasets

On the field of ship detection and fine-grained recognition, several challenges related to datasets have been identified. These challenges include:

1. Limited open source datasets and small data volume: There is a scarcity of open source datasets, particularly for ship fine-grained recognition. The available datasets are often small in size, which limits the amount of training data.
2. Confusing dataset labeling: Existing datasets, such as the HRSC-2016 [35] dataset, suffer from confusing multilevel labeling. This includes missing the labeling of small ships and incorrectly labeled ship classes, which can affect the accuracy of ship recognition algorithms.
3. Imbalance in ship classes and appearance features: Current ship class datasets suffer from an imbalance in the number of ship classes and appearance features. The majority of datasets have a larger number of civilian ships, while warships, which are strategically significant and more challenging to identify due to inter-class similarities and intra-class differences have limited samples.

To address these challenges, we collected samples from various publicly available datasets, including Google Earth, and reclassified and annotated them. This effort resulted in the creation of a new fine-grained warship dataset called FGWC-18. The FGWC-18 dataset includes 18 ship classes, namely Arleigh Burke class; Whitby Island class; Perry class; San Antonio class; Ticonderoga class; Abukuma class; Tarawa class; Austen class; Wasp class; Freedom class; Independence class; Horizon class; Atago class; Maestrale class; Akizuki class; Asagiri class; Kidd class; and Kongo-class. Specific types and the number distribution are shown in Figures 5 and 6. The dataset also includes annotations for ship components, including main guns, vertical launch systems, chimneys, and flight decks. For the convenience of subsequent research, we included ship components in a separate dataset named FGWC-components. As can be observed in Figure 5, our dataset covers most of the common types of warships, and these have greater detection significance and little presence in other datasets. As shown in Figure 6, the distribution of the number of different warship types has a long-tailed distribution, which is widely present in remote sensing

imagery. Smaller ship types are often not detected well. Therefore, the detection results of the smaller ship types are a judgment criterion for the detection algorithm.

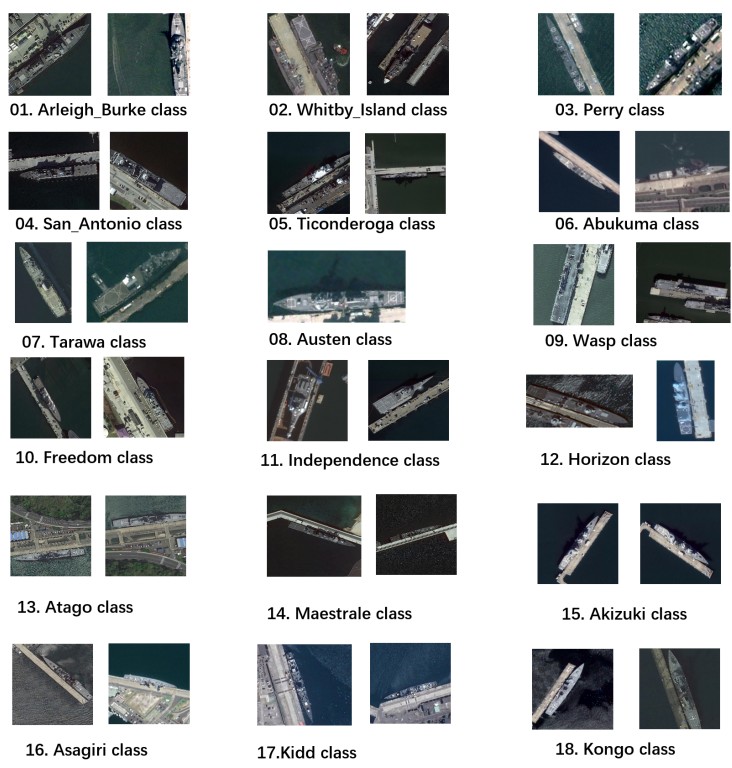

**Figure 5.** Specific examples for FGWC-18 categories.

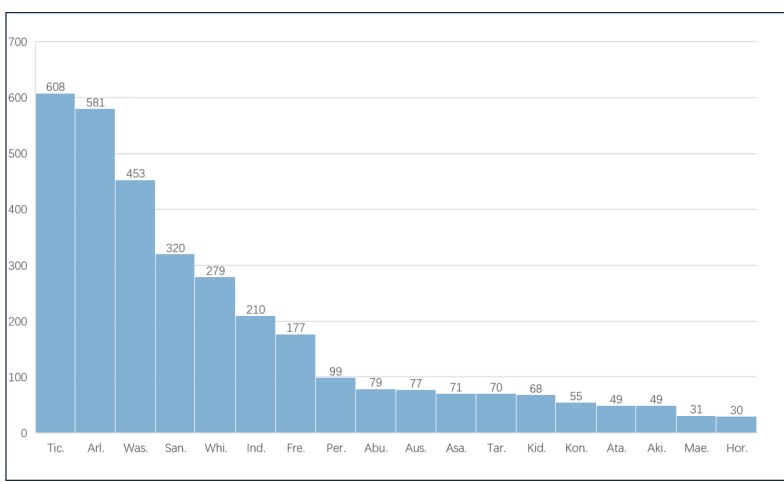

**Figure 6.** Distribution of the number of samples in each category of FGWC-18.

Validation experiments were conducted on the FGWC-18 dataset for ship recognition as a whole and ship components. Additionally, to evaluate the generalization ability and robustness of the proposed algorithm, experimental validation was performed on two ship fine-grained datasets with more labeled categories, namely FGSC-23 [36] and FGSCR-42 [37]. The results of these experiments demonstrate the significant potential of the proposed method in ship detection and fine-grained recognition tasks.

2.4.1. HRSC2016

The HRSC2016 dataset, released by the Northwestern Polytechnical University in 2016, was extracted from six important ports in Google Earth and consists of a total of 1680 images. However, only 1061 of these images were valid for annotation. The dataset was classified at multiple levels, with a total of 28 defined categories. The first level of classification is based on ship types, while the second level provides coarse classification into the four major categories: aircraft carriers, warships, merchant ships, and submarines. The third level of classification further refines the ship models.

This dataset poses several challenges for ship detection and classification tasks. Some of these difficulties include:

1.  Dense arrangement of ships along the shore: Ships are often densely arranged along the shore in the dataset, resulting in a high degree of overlap in the labeling frames. This can make it challenging to accurately detect and classify individual ships.
2.  Complex background in remote sensing images: The background of the remote sensing images in the dataset is complex, and there is a significant degree of similarity between the ships to be detected and the nearby shore textures. This adds to the difficulty of distinguishing ships from the background.
3.  Variation in ship scales: The dataset contains ships of various scales, with a wide range of sizes present in the same image. Detecting and classifying ships with varying scales adds complexity to the task.
4.  Numerous ship categories: The dataset includes dozens of different ship categories, making classification detection a challenging task. Each category may have distinct visual characteristics that need to be learned and recognized.
5.  Within-category ship variations: Each ship category in the dataset contains multiple different ships, which further complicate the classification detection task. The variations within each category require the model to be robust and capable of distinguishing between different ships within the same category.
6.  Small sample size and insufficient learning: The number of ships in each category is not large, leading to a small sample size for training. Insufficient learning and training data can result in reduced model performance and poor robustness.
7.  Cloud masking issues: The dataset may also have challenges related to cloud masking, where clouds obstruct the components of the ships or introduce additional visual noise.

These difficulties and challenges highlight the complexity of ship detection and classification tasks using the HRSC2016 dataset.

2.4.2. FGSC-23

The FGSC-23 dataset consists of images that were primarily derived from publicly available Google Earth data and panchromatic remote sensing images captured by the GF-2 satellite. The images have a resolution ranging from 0.4 to 2 m. This dataset includes a total of 4052 foreign ship slices.

The dataset comprises 23 categories, including 22 ship target categories and 1 non-ship target category that resembles a ship. The ship target categories encompass various types of ships such as container ships, bulk carriers, fishing boats, passenger ships, and more. Each ship category is labeled with a number ranging from 0 to 22. The number of samples for each category may vary, resulting in an imbalanced distribution.

The FGSC-23 dataset exhibits several characteristics:

Category diversity: The dataset provides a fine-grained categorization of ship samples within the images. For instance, the cargo ship category is further subdivided into five fine-grained categories, including container ships, bulk carriers, car carriers, oil tankers, and liquefied gas ships. This level of category diversity allows for more detailed and precise ship recognition.

Image diversity: The FGSC-23 dataset encompasses both panchromatic and synthetic color images. The dataset captures a variety of imaging weather conditions, lighting condi-

tions, ship backgrounds (such as offshore and alongside), and varying ship orientations. Additionally, the dataset features a range of image resolutions between 0.4 and 2 m. The diverse nature of the images, including different backgrounds, lighting conditions, and resolutions, increases the complexity and difficulty of ship recognition tasks. However, this diversity also contributes to training recognition models with better generalization ability, as the models learn to handle various real-world scenarios.

Overall, the FGSC-23 dataset offers a wide range of ship categories and diverse images, providing researchers with valuable data for ship recognition and classification tasks.

### 2.4.3. FGSCR-42

The FGSCR-42 dataset consists of images collected from various sources, including Google Earth and popular remote sensing imagery datasets like DOTA, HRSC2016, and NWPU VHR-10. This dataset focuses on the most common ship types and contains a total of 9320 images.

The images in the FGSCR-42 dataset have varying sizes, ranging from approximately $50 \times 50$ to $1500 \times 1500$ pixels. Different subclasses within the same parent class may exhibit different aspect ratios, leading to significant intraclass variations. This means that ships of the same category can have diverse appearances due to variations in their aspect ratios, further challenging the ship recognition task. On the other hand, interclass variations, or the differences between different ship categories, may be relatively small.

One important aspect of the FGSCR-42 dataset is its representation of real-world scenarios. In reality, the different types of ships can be docked together, resulting in images where multiple ships appear simultaneously. This scenario can introduce additional complexity and potential interference in ship recognition algorithms.

Compared to other fine-grained visual classification (FGVC) datasets, the FGSCR-42 dataset is particularly suitable for fine-grained ship classification. Its inclusion of realistic scene representations, with variations in ship types, aspect ratios, and the presence of multiple ships in an image, provides a challenging and representative dataset for fine-grained ship classification tasks.

By utilizing the FGSCR-42 dataset, researchers can develop and evaluate ship recognition models capable of handling real-world ship classification scenarios.

### 2.5. Evaluation Metrics

In our evaluation of RS ship recognition methods, we primarily utilize the intersection over union (IoU) metric to distinguish detection results. IoU is calculated by dividing the overlap area between the predicted bounding box and the ground truth bounding box by their combined area. By comparing the IoU value against a threshold, we can determine whether a detection is considered a true positive (TP) or a false positive (FP). If a ground truth box does not have a corresponding detection, it is labeled a false negative (FN).

To evaluate the overall performance of the RS ship recognition method, we employ the widely used mean average precision (mAP) as the evaluation metric. mAP is derived from calculating the precision (P) and recall (R) values. Precision is defined as the ratio of true positives to the sum of true positives and false positives (P = TP/(TP + FP)), while recall is defined as the ratio of true positives to the sum of true positives and false negatives (R = TP/(TP + FN)).

mAP is then computed by averaging the precision–recall curves across all categories, and can be expressed as: $\mathrm{mAP} = (1/A) \sum_{a=1}^{A} \int P_a(R_a) dR_a$

Here, A denotes the total number of categories, and $P_a$ and $R_a$ represent the precision and recall for each category *a*, respectively. The mAP value provides an overall assessment of the model's performance across all categories, considering both precision and recall.

## 3. Results

### 3.1. Experimental Parameters

Related-YOLO improved the training of the experiments by setting the batch size to 8 and randomly selecting 8 samples of data from the training set each time for a total of 200 epochs of training.

### 3.2. Experimental Results and Analysis

3.2.1. Ablation Experiment

In this section, we discuss the impact of each component in the network. We choose YOLOV5 as our baseline task. Table 1 shows the experimental results of the ablation study on the FGWC-18 dataset regarding the overall fine-grained recognition of ships. Figures 7 and 8 show the recognition results of several typical scenarios in FGWC-18 for different networks in the ablation study. It can be seen that the modules and algorithms that we introduced for the challenges described in Section 1, respectively, have worked well.

**Table 1.** Experimental results on FGWC-18 using algorithms that generate anchor boxes in different ways for ship type recognition.

| Method | YOLO V5 | YOLO V5 | YOLO V5 | YOLO V5 | YOLO V5 |
|--------|---------|---------|---------|---------|---------|
| | | Hierarchical clustering | Hierarchical clustering | Hierarchical clustering | Hierarchical clustering |
| | | | Multiscale detection layer | Multiscale detection layers | Multiscale detection layer |
| | | | | Deformable convolution | Deformable convolution |
| | | | | | Relational attention module |
| Arl. | 89.5% | 90.1% | 90.7% | 91.3% | 95.5% |
| Whi. | 99.5% | 95.3% | 94.2% | 96.3% | 99.5% |
| Per. | 96.6% | 97.1% | 96.5% | 98.8% | 98.8% |
| San. | 79.2% | 79.5% | 80.1% | 81.3% | 83.5% |
| Tic. | 86.2% | 89.3% | 88.2% | 89.7% | 90.5% |
| Abu. | 97.3% | 95.5% | 96.3% | 97.8% | 98.8% |
| Tar. | 22.2% | 41.6% | 54.7% | 61.0% | 68.4% |
| Aus. | 82.0% | 82.5% | 83.4% | 85.6% | 89.2% |
| Was. | 99.1% | 98.5% | 97.2% | 99.5% | 99.5% |
| Fre. | 92.1% | 93.5% | 94.3% | 94.8% | 95.3% |
| Ind. | 99.5% | 96.5% | 97.1% | 98.9% | 98.6% |
| Hor. | 55.4% | 57.7% | 58.9% | 59.1% | 79.7% |
| Ata. | 65.7% | 70.6% | 72.4% | 73.6% | 84.4% |
| Mae. | 44.8% | 62.5% | 64.2% | 68.9% | 80.5% |
| Aki. | 67.5% | 69.3% | 69.5% | 72.4% | 75.9% |
| Asa. | 99.5% | 98.4% | 99.6% | 98.2% | 99.2% |
| Kid. | 69.3% | 70.5% | 72.4% | 75.5% | 80.1% |
| Kon. | 63.8% | 68.7% | 69.4% | 75.5% | 80.1% |
| mAP@50 | 78.2% | 81.1% | 82.1% | 84.3% | 88.8% |

(1)   Effectiveness of Adaptive Anchor Boxes

In the preceding paragraphs, we propose an adaptive anchor box generation mechanism to enhance the detection ability of the network. They apply a hierarchical clustering algorithm at the input of the network to dynamically generate anchor boxes. The experiments were conducted on the FGWC-18 dataset for ship category detection and the FGWC-components dataset for ship component detection.

For ship category detection on the FGWC-18 dataset, the baseline network achieves a mean average precision (mAP) of 78.2%. When incorporating hierarchical clustering into the baseline, the mAP improves by 2.9%. This indicates that the adaptive anchor box generation mechanism enhances the performance of the network.

To further investigate the working mechanism of the adaptive anchor box, we conduct additional experiments on the FGWC-components dataset. The baseline network achieves an mAP of 49.5% on this dataset. Integrating K-means clustering into the baseline improves the mAP by 2.7%, while replacing K-means with hierarchical clustering leads to a larger improvement of 3.1%. These improvements are mainly attributed to the enhancement in network recall, which increases from 48.9% in the baseline to 52.2% with K-means clustering and 55.3% with hierarchical clustering.

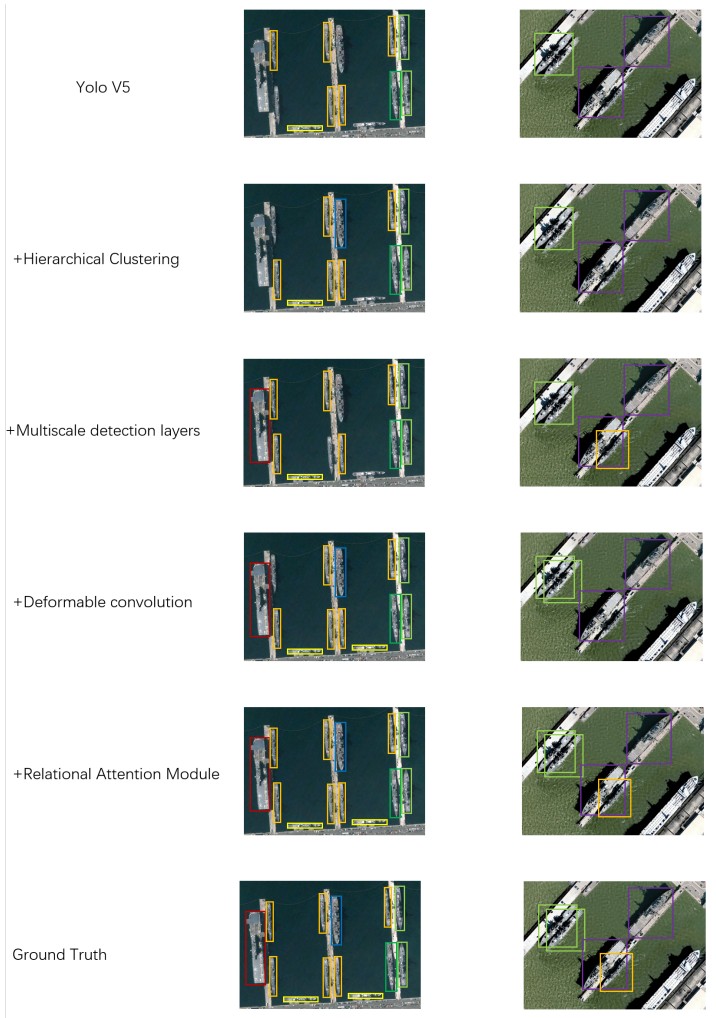

**Figure 7.** Visualization of multi-scenario experimental results in ablation experiments.

The results as shown in Table 2 indicate that the adaptive anchor box mechanism effectively improves the network recall and the detection of small targets, while mitigating the need for manually setting hyperparameters based on strong prior knowledge. The dynamic generation of anchor boxes based on clustering algorithms boosts the performance of ship detection and component recognition tasks.

**Table 2.** Experimental results of different anchor frame generation methods on the FGWC-components dataset.

| Method | YOLO V5 | YOLOV5 + K-Means | YOLO V5 + Hierarchical Clustering + Relational Attention Module |
|---|---|---|---|
| Main gun | 33.9% | 43.8% | 49.2% |
| Vertical launch system | 62.4% | 62.5% | 62.8% |
| Chimney | 19.9% | 30.4% | 39.7% |
| Flight deck | 69.8% | 70.1% | 70.5% |
| mAP@0.5 | 49.5% | 52.2% | 55.3% |

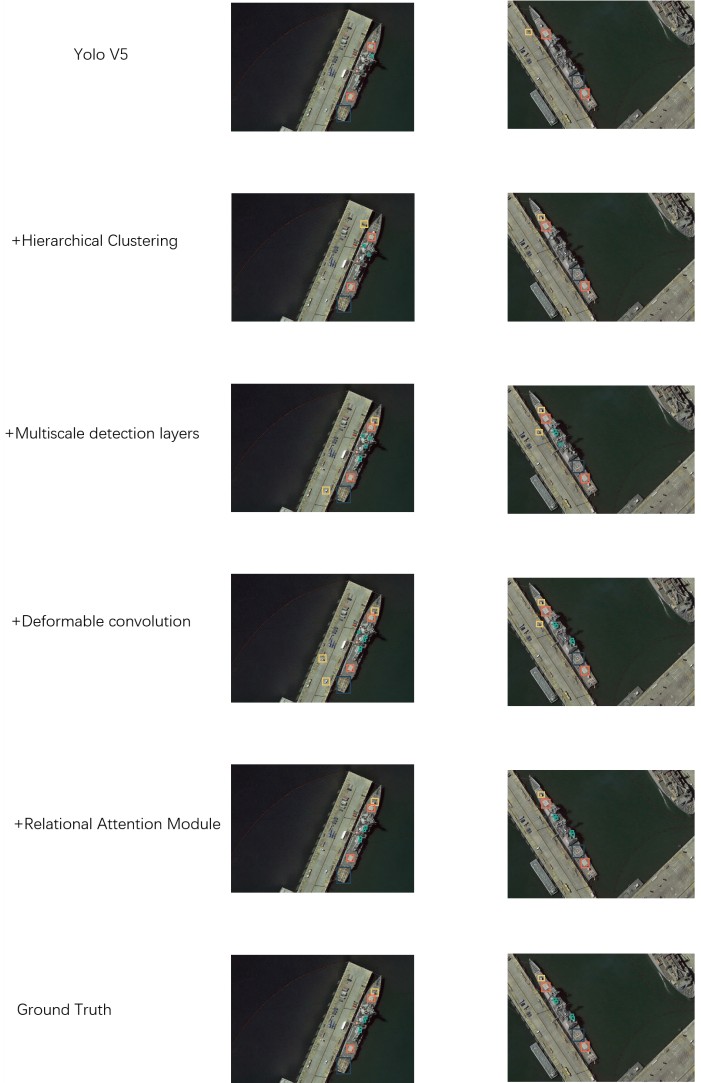

**Figure 8.** Visualization results for ship components.

(2)   Effectiveness of Multiscale Detection Layers

In the feature extraction stage of the backbone, the scale difference between the ship as a whole and the ship components is large; thus, in order to better extract the features, we redesigned the structure of the feature extraction network, added a small target detection head, and changed the original three-scaled structure to the current four-scaled. There is a certain improvement in the detection ability of the network, especially for small target detection. As shown in Table 1, adding the small target detection head in the baseline improves the mAP by 1.0%.

Similarly, we conducted a comparison experiment on FGWC-components and the results are shown in Table 3. As shown in Table 3, the addition of the small target detection

head in the baseline improves the mAP by 2.8%. Again, the performance improvement is mainly seen in some small component categories where the AP boost is significant, such as the main gun and chimney.

(3)  Effectiveness of Deformable Convolution

We utilize deformable convolution to improve the feature extraction of irregular objects, and we conducted experiments on the FGWC-components dataset for ship components. As shown in Table 4, after we add the deformable convolution in the first convolutional layer, the mAP is increased by 6.1%.

**Table 3.** Experimental results of different anchor box generation methods on the FGWC-components dataset.

| Method | YOLO V5 | YOLOV5 + Multiscale Detection Layers |
|---|---|---|
| Main gun | 49.2% | 52.5% |
| Vertical launch system | 62.8% | 63.9% |
| Chimney | 39.7% | 44.7% |
| Flight deck | 70.5% | 71.4% |
| mAP@0.5 | 55.3% | 58.1% |

**Table 4.** Experimental results of deformable convolution at different locations on the FGWC-components dataset.

| Method | Baseline | Layer 1 | Layer 3 | Layer 5 | Layer 7 |
|---|---|---|---|---|---|
| Main gun | 52.5% | 55.8% | 52.1% | 48.2% | 46.3% |
| Vertical launch system | 63.9% | 69.2% | 59.2% | 58.7% | 55.9% |
| Chimney | 44.7% | 54.4% | 48.2% | 46.7% | 42.1% |
| Flight deck | 71.4% | 77.3% | 71.1% | 69.3% | 67.7% |
| mAP@0.5 | 58.1% | 64.2%% | 57.9% | 55.7% | 53% |

In addition to this, we also experimentally verified the components where deformable convolution is added. The experimental results show that the effect of adding in the first layer is the best, and the improvement rate of mAP is 6.1%. From this, we can analyze that our introduction of deformable convolution mainly targets ship components with different shapes, which occupy a relatively small component of the feature map, and the high-level features lose the features of small targets.

(4)  Effectiveness of the Relational Attention Module

The introduction of relational attention aims to view the separated unrelated multi-target recognition tasks as a whole, to utilize the detection results of other easy-to-detect targets to help hard-to-detect targets. It is the focus of this paper.

We conducted experiments on the FGWC-components dataset for ship components. As shown in Table 5, the baseline network obtained an mAP of 64.2%. And, we improved the mAP by 9.9% after adding the relational attention mechanism. In particular, it can be seen that the relational attention mechanism greatly pulls up the overall average, especially for species that were previously difficult to detect.

**Table 5.** Experimental results of the relational attention module on the FGWC-components dataset.

| Method | YOLO V5 | YOLOV5 + the Relational Attention Module |
|---|---|---|
| Main gun | 55.8% | 69.5% |
| Vertical launch system | 69.2% | 75.4% |
| Chimney | 54.4% | 68.3% |
| Flight deck | 77.3% | 83.2% |
| mAP@0.5 | 64.2% | 74.1% |

3.2.2. Comparison with the State of the Art

In this section, we conducted a performance comparison between our proposed Relead-YOLO and seven state-of-the-art recognition approaches, namely R2CNN, RRPN [38],

SCRDet, R3Det, CSL [39], ROI Transformer, and ReDet. Among these approaches, R2CNN and RRPN are general-oriented object recognition networks, while the rest are specifically designed for remote sensing object recognition.

To provide a comprehensive evaluation, all methods were implemented on the FGWC-18 dataset. The performance, measured in terms of mean average precision (mAP), for each model on FGWC-18, is presented in Table 6.

**Table 6.** Experimental results of each method on the FGWC-18 dataset.

| Method | Ours | R2CNN | RRPN | SCRDet | R3Det | ReDet | CSL | ROI Transformer |
|--------|------|-------|------|--------|-------|-------|-----|-----------------|
| Arl. | 95.5% | 87.1% | 88.4% | 84.6% | 84.9% | 88.1% | 74.6% | 84.9% |
| Whi. | 99.5% | 63.9% | 67.1% | 63.9% | 53.5% | 72.7% | 60.9% | 78.5% |
| Per. | 98.8% | 70.8% | 75.2% | 80.4% | 83.6% | 88.0% | 80.6% | 84.9% |
| San. | 83.5% | 87.7% | 86.7% | 57.7% | 67.3% | 86.4% | 77.2% | 88.9% |
| Tic. | 90.5% | 87.8% | 86.2% | 81.2% | 87.1% | 90.6% | 87.7% | 89.7% |
| Abu. | 98.8% | 95.5% | 96.3% | 94.7% | 92.1% | 93.6% | 97.8% | 90.1% |
| Tar. | 68.4% | 80.5% | 98.6% | 94.7% | 88.6% | 99.7% | 99.1% | 87.2% |
| Aus. | 89.2% | 80.9% | 85.1% | 82% | 82% | 88.4% | 84.6% | 89.5% |
| Was. | 99.5% | 80.1% | 82.5% | 83.7% | 84.2% | 86.5% | 86.4% | 87.2% |
| Fre. | 95.3% | 82.4% | 89.3% | 82.5% | 84.3% | 87.2% | 82.1% | 83.3% |
| Ind. | 98.6% | 87.1% | 88.5% | 84.3% | 89.7% | 89.6% | 90.1% | 92.2% |
| Hor. | 79.7% | 75.6% | 73.2% | 72.1% | 78.8% | 72.1% | 73.2% | 74.1% |
| Ata. | 84.4% | 69.9% | 70.1% | 68.2% | 66.8% | 63.5% | 75.4% | 78.6% |
| Mae. | 80.5% | 72.1% | 70.2% | 73.6% | 77.6% | 67.8% | 79.5% | 80.0% |
| Aki. | 75.9% | 70.3% | 70.5% | 74.1% | 73.4% | 79.2% | 84.2% | 82.1% |
| Asa. | 99.2% | 50.2% | 20.3% | 53.1% | 56.2% | 49.3% | 44.8% | 60.7% |
| Kid. | 80.1% | 80.6% | 84.1% | 85.5% | 86.3% | 89.7% | 80.5% | 83.1% |
| Kon. | 80.1% | 76.8% | 70.1% | 53.2% | 40.1% | 67.4% | 70.9% | 68.3% |
| mAP@50 | 88.8% | 78.9% | 77.9% | 77.8% | 78.7% | 83.3% | 82.2% | 83.5% |

From the results, it is evident that our Relead-YOLO achieves state-of-the-art performance on FGWC-18, outperforming the other approaches in terms of mAP. This highlights the effectiveness and superiority of our proposed method in remote sensing object recognition tasks.

### 3.2.3. Robustness Test

To provide a more detailed analysis of the algorithm's performance, experiments were conducted on two datasets: FGSC-23 and FGSCR-42. The FGSC-23 dataset consists of 3596 optical remote sensing images, encompassing 22 types of ships. The images are in JPG format, with a resolution ranging from 0.4 m to 2 m, and pixel sizes ranging from 40 to 800.

Since the FGSC-23 and FGSCR-42 datasets do not provide component information, the Related-YOLO algorithm was first trained on the FGWC-components dataset to predict ship components. The obtained component prediction results were then added to the annotation information of the datasets before conducting the experiments.

The experimental results, as shown in Table 7, indicate that the proposed method outperforms other classification methods on different remote sensing ship datasets. This demonstrates good classification effects and exhibits high robustness in classifying remote sensing ship images.

**Table 7.** Experimental results of each method on the FGSCR-42 and FGSC-23.

| Method | Ours | R2CNN | RRPN | SCRDet | R3Det | ReDet | CSL | ROI Transformer |
|--------|------|-------|------|--------|-------|-------|-----|-----------------|
| FGSCR-42 | 95.5% | 77.4% | 87.2% | 88.1% | 89.3% | 91.6% | 92.3% | 93.1% |
| FGSC-23 | 92.1% | 77.9% | 81.7% | 85.9% | 82.1% | 84.2% | 82.6% | 87.8% |

## 4. Discussion

This paper proposes a new network called Related-YOLO to solve the problem of difficult detection of rotating targets and difficult fine-grained detection in remote sensing images. To evaluate the performance of Related-YOLO, we conduct experiments on three datasets: FGWC-18, FGSC-23, and FGSCR-42. The results demonstrate the superiority of our algorithm compared to popular object detectors. On the FGWC-18 dataset, Related-YOLO achieves an impressive mAP of 88.8%. Furthermore, on the FGSC-23 and FGSCR-

42 datasets, Related-YOLO outperforms other methods and achieves a state-of-the-art performance with mAP values of 92.1% and 95.5%, respectively.

Ablation experiments were also conducted to validate the effectiveness of the proposed modules in Related-YOLO. The results of these experiments further confirm the significance of the introduced components in achieving a superior detection performance.

In summary, our method effectively addresses the challenges of detecting rotating targets and performing fine-grained detection in ship detection, which greatly enhances the applicability of remotely sensed images in the marine field. However, it should be noted that our algorithm introduces additional parameters, which slightly reduces the detection speed of the model.

In future work, we aim to design algorithms that make better use of the internal geometric features of ships. The goal is to develop an end-to-end network for ship detection and recognition, which will provide a more balanced trade-off between model performance and speed. By leveraging the internal geometric features of ships, we anticipate achieving improved efficiency in ship detection and recognition tasks. This will further enhance the overall performance of the model while maintaining a reasonable detection speed.

## 5. Conclusions

In this paper, we introduce a novel ship detection framework for optical remote sensing images called Related-YOLO. Unlike traditional ship recognition methods that treat each ship as a whole object, our framework considers individual ship components as detection targets as same as ship and leverages the geometric relationships among components and that between components and the overall ship for ship recognition.

To enforce the geometric constraints between components, we propose a new module called the relational attention mechanism. This mechanism allows the model to effectively capture and utilize the geometric relationships. Additionally, we address the challenges posed by multi-shape and multi-scale components through techniques such as adaptive anchor frames, deformable convolution, and small target detection layers.

Through empirical studies and comparisons, we demonstrate that Related-YOLO is a highly effective ship detection framework that achieves state-of-the-art performance on various datasets.

**Author Contributions:** S.M. and Q.W. proposed the original idea and designed the experiments. S.M. performed the experiments and wrote the manuscript. Q.W. reviewed and edited the manuscript. Z.P. and W.W. revised the manuscript. Y.H. and G.Z. contributed the computational resources and edited the manuscript. All authors have read and agreed to the published version of the manuscript.

**Funding:** This work was supported by the Aerospace Information Research Institute, Chinese Academy of Sciences.

**Data Availability Statement:** Publicly available datasets were analyzed in this study. This data can be found here: [https://github.com/ygc-iecas8/FGWC-18].

**Conflicts of Interest:** The authors declare no conflicts of interest.

## Abbreviations

The following abbreviations are used in this manuscript:

| | |
|---|---|
| CNN | convolutional neural networks |
| HBBs | horizontal bounding boxes |
| OBBs | oriented bounding boxes |
| IoU | Intersection over Union |
| NMS | Non-Maximum Suppression |

| | |
|---|---|
| ROIs | Region of Interests |
| CSL | Circular Smooth Label |
| YOLO | You Only Look Once |
| FGWC | Fine-Grained Warship Classification |
| FGSC | Fine-Grained Ship Classification |
| FGSCR | Fine-Grained Ship Classification in Remote sensing images |
| FGVC | fine-grained visual classification |
| mAP | mean average precision |
| TP | true positive |
| FP | false positive |
| FN | false negative |
| R2CNN | rotational region CNN |
| RRPN | Radar Region Proposal Network |
| SCRDet | small, cluttered and rotated objects Detector |
| R3Det | Refined Single-Stage Detector |
| ReDet | Rotation-equivariant Detector |

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
