# Peer review of "A Recognition Model Incorporating Geometric Relationships of Ship Components"

_remotesensing, doi:10.3390/rs16010130_

Round 1

Reviewer 1 Report

Comments and Suggestions for Authors

Ship recognition using optical remote sensing images finds wide applications in various 1 domains, such as fishery management, ship traffic surveillance, and maritime warfare. This is of great practical significance and value for optimizing the efficiency of ship management and control and providing automation and intelligence of management and control. The content of the manuscript is detailed, the method is clearly explained, and the conclusion analysis is feasible. However, the structure of the manuscript has major problems and needs to be adjusted accordingly. The questions are listed below. It is recommended that the manuscript be published after minor revision.

Section 1 and 2: the two can be combined. Relevant research results should be explained in the introduction, and the chapter arrangement in the manuscript is more like a dissertation than an academic paper. Merger should compress and refine the length.

Section 4.1 explains the study data and should be placed in Section 3.

Section 4.2 explains the study methods and should be placed in Section 3.

It is recommended to increase the discussion and focus on the application prospects and limitations of the research, as well as the focus of the next step.

Reviewer 2 Report

Comments and Suggestions for Authors

In the article titled "A Ship Recognition Model Incorporating Geometric Relationships of Ship components", Shengqin Ma et al. proposed a ship detection model that utilizes geometric relationships of ship components. Basically, the manuscript introduce the main idea, method and experiment clearly. The main contribution of this study in proposing a new idea to solve the fine-grained detection of ship targets. It bring the knowledge into the detection and recognition, which is a novel idea in the field.

While, a few typos is seen in this manuscript. Some details of experiment and method should be more clarified.

Therefore, MINOR revision has to be done before this article could be accepted for publication in the Remote Sensing.

 The following are specific modification comments:

 When introducing the relational attention and adaptive mechanism boxes, it is recommended that the author give specific mathematical formulas so that readers can more clearly understand the principles and implementation of these methods.

In the experiment, it is recommended that the author provide more details of the experimental settings, such as data deletion, network structure and hyperparameter settings, so that other researchers can conduct the experiments and conduct comparative studies.

The proposed method adopt the relationship among components of the ship as a regulation, and utilize the attention module to model it. So is it necessary to pre-build the ship model for this network? Please provide more details.

Comments on the Quality of English Language

The English writing is good.

Reviewer 3 Report

Comments and Suggestions for Authors

Thank you for sharing this article with me. The paper presents a model called Related-YOLO that utilizes the mechanisms of relational attention to stress positional relationships between ship components, extracting key features more accurately. I have the following comment:

- The word "Ship" appears twice in the title, you can easily remove one of them. Please revise the title and make it more concise.

- Please add the value addition of the study (or its beneficiaries) in a line or two to the abstract

- Since there are too many technical jargon (and/or abbreviations), please add a table of abbreviations to the paper. 

- Please provide references for lines 15-21.

- In many instances in the paper, abbreviations have been presented in a way that won't be appropriate for non-technical (or noncomputer science) readers. For example, line 22 presents YOLT, and line 23 presents R2-CNN yet none of these are discussed for the readers to apprehend what they mean. There are many such instances throughout the paper. Please revise the entire paper carefully for such issues. Merely naming different items does not add any scientific value. Rather it is the discussion on the relevant papers and models that progress the science. Please improve the discussions throughout the paper beyond naming various techniques.

- In line 32 the authors refer to 2 aspects however only one is presented and numbered in line 33. The second one appears 2 paragraphs later. Please present (or clarify) both parts together then explain them subsequently. Also, there are no references in this para for the claims made.

- The addition of the study is not clarified in the introduction. Please clarify the value added to the target stakeholders.

- The issue of presenting literature at face value is evident throughout Section 2 (related works). Please elaborate on the various studies you have presented and discuss the key innovations rather than running mentions only.

- Please add a method diagram to the method section that graphically show the steps taken in this study. 

- Section 3.1 and 3.2 have only one reference. These are established techniques yet the authors have not provided proper references. This indicates the lack of consultation of state-of-the-art literature. Please improve the referencing.

- In terms of YOLO as a method, the innovation is not shown in the study. how is your YOLO model different than the models of existing studies for example see the following and discuss how your model is different?

    10.23919/ACES-China60289.2023.10249265

     https://link.springer.com/chapter/10.1007/978-981-99-6755-1_10

    https://www.mdpi.com/2071-1050/15/3/1866

- The discussions on Figures 4 and 5 are very weak. Please discuss the key takeaways of the figures in the text.

- The rationale for the section of the datasets is also not clear in the study. Why were these datasets selected? what makes them the best datasets for this study?

- The limitations and future directions are not clear in the conclusions. Please add them to the conclusion of the paper.

Comments on the Quality of English Language

The paper needs language editing

Round 2

Reviewer 3 Report

Comments and Suggestions for Authors

Thank you for addressing my comments.

Comments on the Quality of English Language

Minor editing needed

Author Response

We would like to thank you for your careful reading, helpful comments, and constructive suggestions, which has significantly improved the presentation of our manuscript. We asked for the help of native speakers to polish the English expression of the entire manuscript. The amendments/supplements to the manuscript are given in red fonts.